

# *Nepenthes maximoides* (Nepenthaceae) a new, critically endangered (possibly extinct) species in Sect. *Alatae* from Luzon, Philippines showing striking pitcher convergence with *N. maxima* (Sect. Regiae) of Indonesia

Charles King and Martin Cheek

Science, Royal Botanic Gardens, Kew, Richmond, UK

## ABSTRACT

*Nepenthes maximoides* sp. nov. (Sect. *Alatae*) is described and assessed as Critically Endangered (Possibly Extinct) from Luzon, Philippines and appears unrecorded in 110 years. The spectacular, large, narrowly funnel-shaped upper pitchers, lids with recurved basal and filiform apical appendages, unlike any other species in the Philippines, closely resemble those of *N. maxima* (Sect. *Regiae*) of Sulawesi–New Guinea, likely due to convergent evolution. Following recent phylogenomic analysis, sect. *Alatae* is divided into two, Sect. *Alatae sensu stricto* of Luzon to Sibuyan (including *N. maximoides*), and Sect. *Micramphorae*, expanded and recircumscribed to encompass those species of the southern Visayas, and Mindanao. A key is provided to the six species now recognised in the newly narrowly recircumscribed Sect. *Alatae*. The number of *Nepenthes* species recorded from Luzon has increased from two in 2001, to eight in 2020, all but one of which are endemic to that island, and four of which appear to be point endemics.

## INTRODUCTION

This paper is one in a series leading to a monograph of the genus *Nepenthes* building on a skeletal revision of the genus (*Jebb & Cheek, 1997*) and the account for Flora Malesiana (*Cheek & Jebb, 2001*). While in 2001 only 85 species were accepted for the genus, today the figure lies at 181 (*Murphy et al., 2020*). In 2001 the geographic units with highest *Nepenthes* species diversity were Sumatra and Borneo, each with over 30 species, while just 12 species were recorded for the Philippines (*Cheek & Jebb, 2001*). However, since then the Philippines has dominated for discovery of new species. In 2013 alone, twelve new species were published from the Philippines (*Cheek & Jebb, 2013a*, *2013b*, *2013c*, *2013d*, *2013e*, *2013f*, *2013g*; *Micheler et al., 2013*), and new species continue to be added every year (*Cheek & Jebb, 2014*; *Cheek, Tandang & Pelser, 2015*; *Gronemeyer et al., 2016*;

Corresponding author
Martin Cheek, m.cheek@kew.org

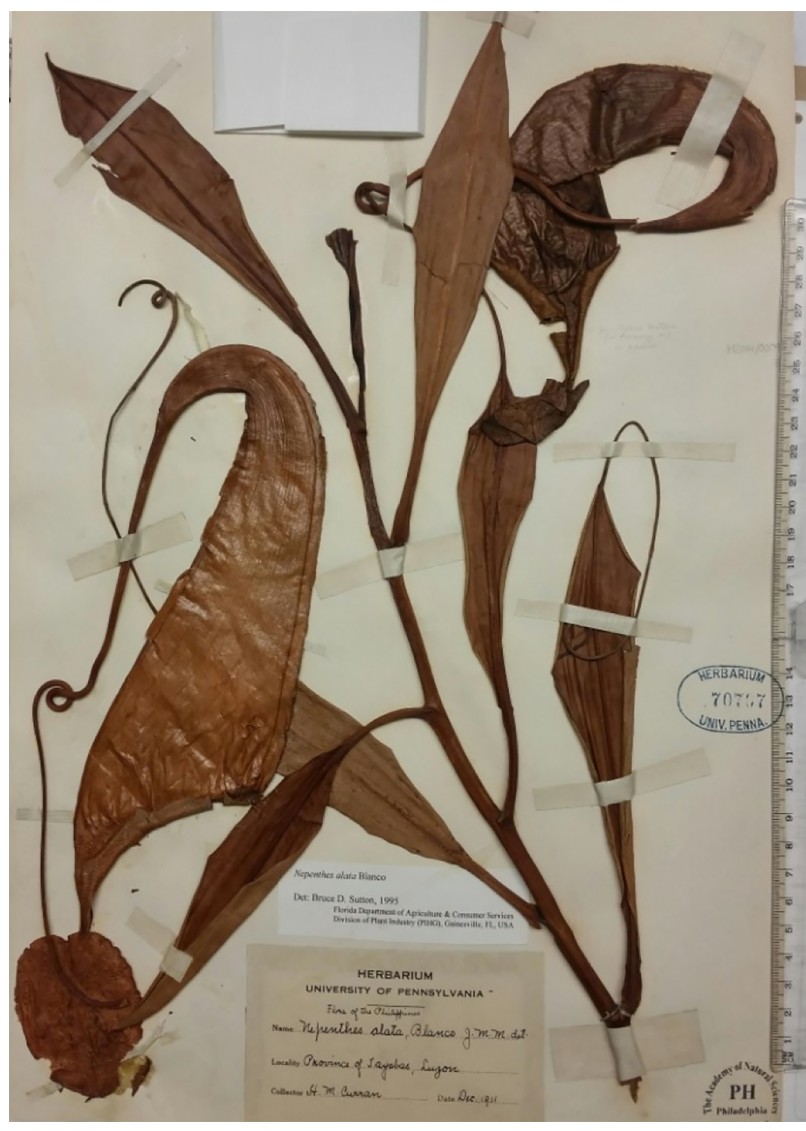

**Figure 1 *Nepenthes maximoides* Cheek.** Photo of the type specimen *Curran s.n.* (Univ. Pennsylvania sheet 70707), PH. Note that material is mounted pitchers facing downward. Photo by Martin Cheek.

*Lagunday et al., 2017*; *Amoroso et al., 2018*; *Robinson, Zamudio & Caballero, 2019a*). The current total of *Nepenthes* species for the Philippines is 59 (*Pelser, Barcelona & Nickrent, 2011* onwards, accessed May 2020) now far exceeding the totals for other phytogeographic units such as Sumatra and Borneo.

However, new species of *Nepenthes* continue to be discovered elsewhere in SE Asia, from Indonesia, with new species described from Halmahera (*Cheek, 2015*), Sulawesi (*Cheek & Jebb, 2016a*, *2016b*) and New Guinea (*Cheek et al., 2018*), and also from Malaysia, with new species from Borneo (*Robinson et al., 2019b*; *Golos et al., 2020*).

During a study of specimens of *Nepenthes* from the Philippines, one specimen, *Curran s.n.* (Fig. 1), previously determined as *N. alata* Blanco (in fact *N. graciliflora* Elmer (*Cheek & Jebb, 2013g*)), was initially set aside since it seemed to have been mislabelled as

from Luzon when in fact the specimen appeared to be *Nepenthes maxima* Nees, a widespread and variable species occurring from Sulawesi to New Guinea but absent from Philippines (*Cheek & Jebb, 2001*). *Nepenthes maxima* is distinctive for its narrowly funnel-shaped upper pitchers, flattened peristome with undulate outer margin, large, ovate-elliptic lid, a usually hooked basal appendage and a filiform apical lid appendage (*Cheek & Jebb, 2001*). No known species in the Philippines remotely resembles it. *Nepenthes maxima* (Borneo-New Guinea) is placed in Sect. *Regiae* Danser which is entirely absent from Philippines (*Cheek & Jebb, 2015*). However, a second inspection of *Curran s.n.* showed that features of the stem and petiole, in contrast to the pitchers, are not those of sect. *Regiae* Danser (*Danser, 1928*), but instead are consistent with those of Sect. *Alatae* Jebb & Cheek, which as currently defined is endemic to the Philippines, occurring from Luzon to Mindanao (*Cheek & Jebb, 2015*). *Curran s.n.* has petioles with conspicuous, patent wings (T-shaped in transverse section) while those of Sect. *Regiae* are inconspicuous or erect (U-shaped in transverse section); the axillary buds are inconspicuous and the indumentum of stem and leaf-blades is absent or inconspicuous (in Sect. *Regiae* the axillary buds are conspicuous, several mm long, and spike-like; and the indumentum is robust, branched, brown and present at least at the stem apex). In Sect. *Alatae* moreover, the phyllotaxy is spiral and not, as in *Regiae*, distichous (*Danser, 1928*; *Cheek & Jebb, 2015*). It seems clear that *Curran s.n.* might have been correctly labelled as a Philippine species, after all.

In this article we key out *Curran s.n.*, distinguishing it from all other species of Sect. *Alatae* in northern Philippines (Luzon and the northern Visayas) describing it as *Nepenthes maximoides* Cheek, diagnosing it from *N. graciliflora* which it had previously been identified as. We also compare it with *N. copelandii* Macfarl. of Mindanao which unusually also shares the narrowly funneliform upper pitcher shape. We present biographical notes on the collector and deduce the location of the geographic source of his collection, likely Mount Banahaw, and consider the likelihood of the extinction of this species. We consider the presence of terminal lid appendages in the genus and their homology; we also discuss morphological convergence with *N. maxima*. Finally, we discuss the division and new, narrow recircumscription of Sect. *Alatae* based on new phylogenomic data.

## MATERIALS AND METHODS

The specimen on which this paper is centred is on loan from PH and was compared with material presently at K including that on loan from A, BISH, BRIT, CAS, GH, L, NY, US and that studied at BO by the second author.

The electronic version of this article in Portable Document Format (PDF) will represent a published work according to the International Code of Nomenclature for algae, fungi and plants (ICN), and hence the new names contained in the electronic version are effectively published under that Code from the electronic edition alone. In addition, new names contained in this work which have been issued with identifiers by IPNI will eventually be made available to the Global Names Index. The IPNI LSIDs can be resolved and the associated information viewed through any standard web browser by appending

the LSID contained in this publication to the prefix http://ipni.org/. The online version of this work is archived and available from the following digital repositories: PeerJ, PubMed Central, and CLOCKSS.

Herbarium citations follow Index Herbariorum (*Thiers, 2020*) and binomial authorities the International Plant Names Index (*IPNI, 2020*). The conservation assessment was made using the categories and criteria of *IUCN (2012)*. Herbarium material was examined with a Leica Wild M8 dissecting binocular microscope fitted with an eyepiece graticule measuring in units of 0.025 mm at maximum magnification. The drawing was made with the same equipment using Leica 308700 camera lucida attachment. The specimen was partly unmounted to expose the characters needed to first identify and second characterise the species. It had originally been mounted with the lower surface of the lid face down, obscuring critical features. The map was made using SimpleMappr (https://www.simplemappr.net).

Information on the likely location of the collection was gathered by analysing other herbarium specimens collected by H.M. Curran recorded on *JStor Global Plants (2020)*.

# RESULTS

## TAXONOMIC TREATMENT

### Key to the species of *Nepenthes* sect. *Alatae* in Luzon and northern Visayas

1 Monopodial erect shrublets 30 cm tall (including inflorescence), climbing stems and upper pitchers always absent; lower pitchers with column extending onto lower surface of the lid; lid lower surface lacking appendages. Sibuyan . . . . . . . **N. argentii** Jebb & Cheek
1 Climbers exceeding 1 m tall (where known), climbing stems and upper pitchers present at maturity; pitchers with column usually absent, if present, not extending onto lower surface of the lid; lid lower surface with basal appendage (except *N. armin*) . . . . . . . . . 2
2 Upper pitchers narrowly infundibulate, broadest at the peristome; lower surface of lid with both basal and apical appendage. Luzon . . . . . . . . . . . . . . . .**N. maximoides** Cheek
2 Upper pitchers ovoid-cylindric, broadest in the basal half; lower surface of the lid lacking an apical appendage (and in *N. armin* a basal appendage also) . . . . . . . . . . . . . . . . . . . 3
3 Upper pitchers lacking convex appendages on their lower surface, outer edge of peristome shallowly lobed; stem angular; male pedicels 3–4.5 mm long. Sibuyan . . . . . . . . . . . . . . . . . . . . . . . . . . . . . . . . . . . . . . . . . . . . . . . . . . . . . . . . . **N. armin** Jebb & Cheek
3 Upper pitchers with basal appendages on their lower surface; outer edge of peristome entire; stem terete; male pedicels >10 mm long . . . . . . . . . . . . . . . . . . . . . . . . . . . . . . 4
4 Stems glabrous, rarely glabrescent; upper pitcher lacking fringed wings; outer surface <20% covered covered in red stellate hairs or lacking them entirely . . . . . . . . . . . . . . . 5
4 Stems persistently pubescent; upper pitchers with fringed wings in upper part; outer pitcher surface >50% covered in grey stellate hairs. Northern Luzon . . . .**N. alata** Blanco
5 Upper pitcher subcylindrical, outer surface 10–15% covered in minute red stellate hairs; lowland coastal ultramafic scrub of N & E Luzon . . . . . . . . . . . . **N. ultra** Jebb & Cheek

5 Upper pitcher with ellipsoid base constricted abruptly to the narrow, cylindrical upper 2/3. Outer pitcher surface lacking stellate hairs. Submontane forest of southern Luzon to Mindanao . . . . . . . . . . . . . . . . . . . . . . . . . . . . . . . . . . . . . .*N. graciliflora* Elmer

**Nepenthes maximoides** Cheek, sp. nov. - Fig. 1, Fig. 2
Differing from *Nepenthes graciliflora* Elmer in the upper pitchers narrowly infundibulate, widest in the distal half at the peristome (not ovoid-cylindric, widest in the proximal half), the peristome broad, flattened, and lobed on the outer edge (not narrowly cylindrical and entire on the outer edge), the lid with an asymmetrically hooked basal appendage and a filiform apical appendage (not symmetrical non-hooked, and absent, respectively). - Type: *Curran s.n.*, Herb. Univ. Pennsylvania sheet number 70707, Academy of Natural Sciences Philadelphia sheet number 01113309 (holotype PH; isotype PNH destroyed, not seen), Philippines, Luzon, 'Tayabas Province' (deduced to be Mt Banahaw, Quezon Prov.) st. December 1911.

*Etymology*. Meaning that the species looks like *Nepenthes maxima* Nees (since it looks so similar to this species that it was confused with it).

*Terrestrial climber* (probably), height unknown. *Rosette* and *short* stems unknown. Climbing stem rounded 8–10 mm diam; phyllotaxy spiralled; internodes 3–6 cm long; axillary buds not conspicuous; indumentum glabrous. *Leaves* petiolate, blades elliptic-oblong (10–)12.5–14 cm by 2.75–3.8 cm wide; apex acute; base gradually decurrent to the petiole; longitudinal nerves arising in the basal 2–2.5 cm of the blade, (2–)3 on each side of the midrib in the outer quarter of the blade, indistinct, pennate nerves more or less patent, indistinct; abaxial surface with sessile depressed-globose glands 0.02–0.03 mm diam., c. 10 per $mm^2$ (Fig. 2M); midrib when young densely, c. 30–40% covered in erect simple hairs 0.1–0.2 mm long (Fig. 2L), margin of young-leaf blade densely hairy, hairs 1–3-armed, patent, acute, 0.06–0.25 mm long (Fig. 2N). *Petiole* clasping the stem by 2/5–½ its circumference, not decurrent, winged, T-shaped in section (2–)3–4 cm long, wings patent, each 2–3 mm wide, glabrous. *Lower and intermediate pitchers unknown. Upper pitchers* (coiled tendril, lid facing away from tendril), narrowly infundibulate or infundibulate-cylindrical in outline, 14–22.5 cm in length; 2.5–3 cm wide at base widening to 4–6.5 cm wide below the peristome, fringed wings reduced to a pair of low ridges; outer surface of pitcher glossy, drying yellowish brown, subglabrous, indumentum extremely sparse, minute and inconspicuous, hairs white, bifid, 0.05 mm long, arms spreading, apices rounded. *Mouth* ovate, slightly concave, at 45 degrees from the horizontal, forming a column below the lid, inner surface of pitcher glandular, immediately below column waxy; peristome rounded or flattened, inverted U-shaped in transverse section (Fig. 1K), 6–6.5 mm wide, c. 8 mm deep, ridges 0.5–0.8 mm apart, developed as acute low ridges 0.1–(0.2) mm high (Fig. 2I), outer edge with 0–2(–3) shallow lobes (Fig. 2A), recurved and rolled (Fig. 2K), inner edge flat, held parallel to pitcher wall, the distal part with teeth inconspicuous, triangular, <0.2 mm long (Fig. 2J); column triangular, 0.8 cm by 0.5 cm (Fig. 2B). Lid elliptic or ovate-elliptic 4.8–5.8 cm by

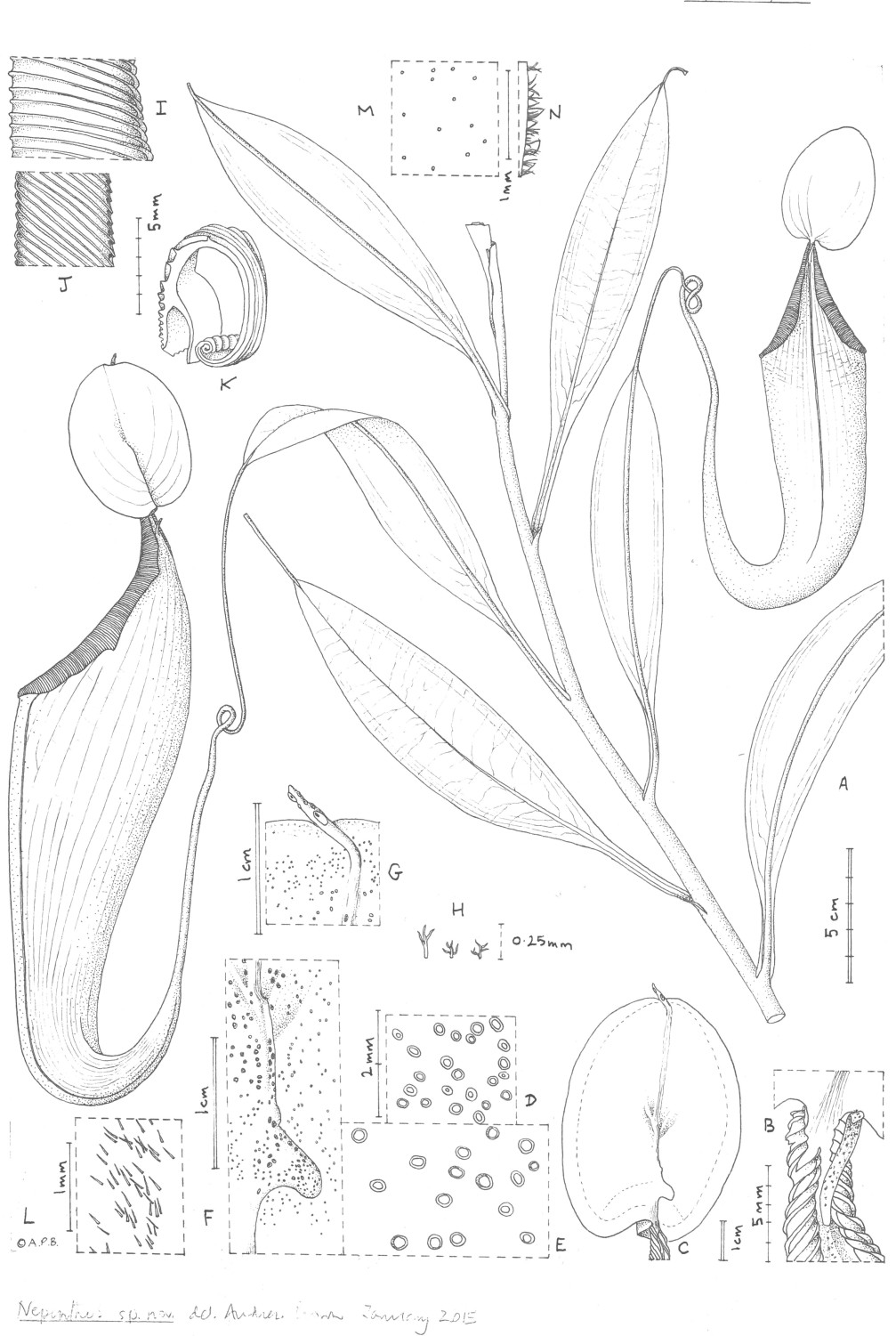

**Figure 2 Nepenthes maximoides.** (A) Habit, showing climbing stem with two upper pitchers; (B) exterior of upper pitcher showing junction of peristome, lid and spur.; (C) lower surface of the lid of the upper pitcher showing basal and apical appendages; dotted line indicates outer limit of nectar glands which are absent from the lid margin; (D) detail outermost lid nectar glands on underside lid; (E) detail
**Figure 2 (continued)**
lid nectar glands near/on basal appendage (same scale as D); (F) detail basal appendage with gland distribution; (G) detail apical appendage; (H) minute branched and stellate hairs of lid margin; (I) peristome near lid, viewed from exterior; (J) inner edge of peristome near lid; (K) peristome of upper pitcher, transverse section, outer surface of pitcher to the right; (L) midrib of leaf blade, adaxial surface, showing simple hairs; (M) sessile depressed-globose glands on abaxial leaf-blade surface; (N) margin of young leaf-blade showing hairs (all: *Curran s.n.* Univ. Pennsylvania sheet 70707). Drawn by Andrew Brown.                                                                               

3.5–4.5 cm, apex broadly rounded, base shallowly cordate, upper surface glabrous apart from sparse sessile red glands c. 0.05 mm diam.; lower surface densely covered (apart from the marginal 4–5 mm band) in monomorphic orbicular to slightly elliptic crater-like nectar glands, nectar glands slightly larger around base of lid, (0.15–)0.25(–0.3) mm diameter (Fig. 2E), gradually becoming smaller towards the lid margin, 0.15(–0.2) mm long (Fig. 2D); both a basal and apical appendage arising from the midline ridge, basal appendage slightly hook-shaped, directed towards base of lid, (Fig. 2F) arising at c. 45 degrees from midline ridge, laterally flattened, oblong-arcuate, 7 mm high and 4–5 mm wide, apex rounded, 1–3 nectar glands per mm$^2$; apical appendage filiform-cylindrical 9.5 by 0.8 mm, arising from midline 4–5 mm from lid apex, nectar glands present on midline; marginal band of lid 3–4(–5) mm wide, lacking nectar glands and with sessile red glands c. 0.05 mm diam., and in the outermost mm, short, erect, 3–5-branched hairs 125–250 µm long and broad. *Spur* (Fig. 1B) inserted c. 4 mm below lid insertion, straight, erect, simple, 6–6.2 mm long, apex shortly bifid, surface moderately densely covered in erect 2–3-armed brown hairs, 0.05–0.08 mm long. *Male*, *female inflorescence*, *infructescence* and *seeds* all unknown.

Conservation—if our deduction is correct that the only known specimen of *Nepenthes maximoides* derives from Mt Banahaw (see "Discussion" below), then there is yet some hope that the species might yet have survived extinction since the mountain and its forest are regarded as sacred by the local population. It also has a high level of formal, government protection, designated as a 'Protected Landscape' since 2003. Nonetheless, although its formal protection has increased since it was designated as a forest reserve in 1921, upgraded to National Park in 1941, at each stage the area has been reduced— probably reflecting the steady clearance of its forest upslope for rice cultivation, which can be seen to this day on Google Earth (viewed Feb. 2020). *Barcelona, Pelser & Cajano (2007)* note that the area of Mount Banahaw has previously been affected by large-scale human disturbance, predominantly from pilgrims (it is a holy mountain), tourists and mountaineering groups. According to *BirdLife International (2020)*, 300,000 pilgrims and hikers per annum visit the mountain, using four trails, resulting in habitat degradation. This activity which occurred for many years prior to Mount Banahaw becoming a protected landscape area in 2004, together with illegal logging, and quarrying at two sites (*BirdLife International, 2020*) may have significantly reduced the population of *N. maximoides* within this area. Should our deduction be incorrect and the location for *N. maximoides* be elsewhere in the province, the outlook for its survival is more negative.

*Myers et al. (2000)* estimated that remaining primary vegetation amounts to only 3%, and *Sohmer & Davis (2007)* estimate extinction levels due to habitat destruction as between 9 and 28% in one representative, mainly forest genus, *Psychotria* L.

We here assess *N. maximoides* as Critically Endangered (Possibly Extinct) CR B2ab(iii), since only a single location, represented by a single specimen, is known, with threats, namely habitat degradation and destruction as reported. The area of occupancy is designated as 4 km$^2$ to comply with *IUCN (2012)* directions. The threats referred to above are ongoing: even at Mt Banahaw a forest fire was recently reported to have destroyed 50 Ha of the mountain's forest (*Ranada, 2014*). While one can hope that this species is simply under collected and will be refound at Banahaw and other sites where the required habitat survives (as was the case in Mindanao with *Nepenthes robcantleyi Cheek (2011)*), this seem unlikely. *Nepenthes* are among the most charismatic plant groups in the Philippines and numerous citizen scientists and botanists have targeted them for study, resulting in numerous exciting new discoveries in the last two decades. Given this, it seems improbable that *N. maximoides*, the largest and most spectacular *Nepenthes* in Luzon, the most heavily populated island of the Philippines should not have been refound in 110 years if it actually survives. Should the species be refound, measures should be taken to determine the size of the population and the state of regeneration, and seed collected as a priority to enable mass in vitro propagation for the *Nepenthes* collector market in order to reduce the certain pressure of poaching that would otherwise result from rediscovery.

## DISCUSSION

The top set of this collection was probably deposited at PNH but, with the entire herbarium, destroyed there with much of the city in February 1945 during the battle of Manila between the forces of Japan and USA (*Scott, 2019*). Studies of specimens either requested on loan from or studied during visits at USA herbaria known to have Philippine specimens, namely AA, BISH, BRIT, CAS, GH, NY, PH, US, have failed to find either additional duplicates, or additional collections of this species. *Curran s.n.* had been determined (undated) as *Nepenthes alata* Blanco by Macfarlane, the last monographer of the genus. He had a broad concept of this species and included within it *N. graciliflora*, and other species, which are now widely recognised (*Macfarlane, 1908*). Macfarlane's determination was adopted by Sutton in 1995. However, both those botanists seem to have only looked at this specimen superficially because the original mountings had remained in place. Because the pitchers are mounted face down, it is only by demounting the specimen that most of the diagnostic characters, such as the lid appendages, can be seen.

With the description in this paper of *Nepenthes maximoides*, the number of *Nepenthes* species accepted from Luzon has increased to eight, from two in 2001 (*Cheek & Jebb, 2001*). All but one of these species are endemic to that island, and four, including *Nepenthes maximoides*, appear to be point endemics on current evidence.

Within the Philippines, *Nepenthes maximoides* is most similar in pitcher shape to *N. copelandii* of Mt Apo, Mindanao. This is because both species have narrowly infundibulate upper pitchers, an unusual character in the Philippines. However, *N. copelandii* has dimorphic nectar glands on the lower surface of the lid, 2-flowered partial-peduncles and hairy stems (not monomorphic and glabrous respectively), also the basal lid appendage of the upper pitchers is only obtuse and is not hooked, and not asymmetric, nor is there an apical appendage. Finally, the peristome of the upper pitcher in *N. copelandii* is not flattened, and lobed on its outer edge as in *N. maximoides*, and the inner edge is minutely toothed, not with teeth inconspicuous as in *N. maximoides*.

**Hugh McCollum (or McCullam or McCullom) Curran (1875–1960) and his Philippine collections**

The collector of the type and only known specimen of *Nepenthes maximoides* is stated on the printed specimen label to be H.M. Curran. Searching under 'Curran' on the IPNI (continuously updated) database gives only two people of this name, both American, only one of which appears to have been active in the Philippines. This is Hugh McCollum Curran (1875–1960) listed on IPNI as a collector, not an author. The other, active in USA, was Mary Katharine Curran (1844–1920). Hugh Curran was a prolific collector and a search of IPNI (continuously updated) produced 84 species names with the epithet 'curranii' most of which are from the Philippines and all of which appear to commemorate H.M. Curran, probably because he collected the type specimen. According to his Wikipedia page (*Wikipedia contributors (2020)*, continuously updated) he was trained in N. Carolina and then at Yale Universities before spending 1906–1912 in the Philippines as a forester with the Forestry Bureau of Manila. Thereafter he visited South America for several years but returned to Philippines 1929–41 as Professor of Forestry, survived internment there in the second world war, afterwards returning to Venezuela.

Analysis of specimens under the name Curran on *JStor Global Plants (2020)* which is dominated by type specimens, returns 1,087 items, about half of which are specimens collected by H.M. Curran, and most of these, 534, are from Philippines. H.M. Curran collected specimens under the Forest Bureau of Manila series. His earliest collections, from 1906, are from Palawan and include *Curran* 3891, the holotype of *Nepenthes deaniana* Macf. (destroyed with the Manila herbarium, PNH in 1945), and the lectotype of *N. philippinensis* Macf., *Curran* 3896 (*Cheek & Jebb, 1999*). While his Palawan collections often had notes giving written descriptions of the specimens with altitudes and locations (e.g. *Curran* 3473) those collected in subsequent years tended to lack these details and to consist of only printed labels, often only with the province, month and year: as for the specimen of *N. maximoides*. However, sometimes the name of a mountain or settlement might be given. It is unusual that no Forest Bureau number is assigned to a specimen as in the case of this type specimen. We speculate that this might be because the specimen is sterile. Even today, in the 21st century, collecting sterile specimens for herbaria is often considered dubious practice and such specimens, collected because they excite interest in the collector, are often sadly left unnumbered (M. Cheek pers. obs. 1991–2020).

**Table 1 Mountains with summits above 1,000 m high.** Found in the Quezon and Aurora Provinces (together previously known as Tabayas Province) of Luzon. Most mountains are found on the border with adjoining provinces.

| Name | Altitude (m) | Provinces | Georeference (dms) |
| --- | --- | --- | --- |
| Mount Anacuao | 1,852 | Aurora, Quezon | 16°15′14″N 121°53′22″E |
| Mingan Mountains | 1,901 | Aurora | 15°25′27″N 121°24′25″E |
| Mount Caladang | 1,465 | Laguna, Rizal, Quezon | 14°49′00″N 121°21′00″E |
| Mount Malabito | 1,360 | Laguna, Quezon | 14°41′07″N 121°26′12″E |
| Mount Binangonan | 1,091 | Quezon | 14°38′00″N 121°33′00″E |
| Banahao de Lucban | 1,874 | Laguna, Quezon | 14°04′00″N 121°30′00″E |
| Mount Banahaw | 2,177 | Laguna, Quezon | 14°04′03″N 121°29′32″E |
| Mount San Cristobal | 1,470 | Laguna, Quezon | 14°03′52″N 121°25′36″E |

Such sterile specimens are often collected only as unicates, when the collector might otherwise have a system of collecting large sets. This would help explain why only a single sheet is known of this specimen. However, in any case, Curran usually appeared to collect only small sets—inspection of the JStor Global Plants data (continuously updated), which details herbaria at which duplicates of type specimens collected by him are known, suggests it is rare that more than two duplicates are known of any of his specimens outside of the Philippines. *Van Steenis-Kruseman & Van Steenis (1950*: 123–124*)* state that of Curran's collections 'Many plants in Herb. Manila, at least partially numbered in the F.B. series' yet this may have been written before the destruction in 1945 or in ignorance of it. Today none are known to survive there.

Tayabas Province, given as location of the type specimen collected by Curran (no further details are given), has been renamed and divided into two: Aurora Province to the north, along the eastern coast of Luzon comprising the narrow coastal plain and the seaward edge of the N–S Sierra Madre range, and in the south Quezon Province—the S.E. corner of Luzon and an adjoining part of the Bicol Peninsula. We can only deduce where within this range the type specimen of *N. maximoides* might have been collected. We can be moderately certain that is was in forest on a mountain above 1000 m. alt., since all but one of the other seven species of *Nepenthes* on Luzon occur in such habitats (*Cheek & Jebb, 2013a*, *2013c*, *2013d*, *2013g*; *Cheek, Tandang & Pelser, 2015*). The exception is *Nepenthes ultra* Jebb & Cheek, restricted to ultramafic habitat just above sea-level (*Cheek & Jebb, 2013h*).

Eight mountains in Quezon and Aurora (former Tayabas) meet this specification (see Table 1, Fig. 3). Most of these (six of the eight) are clustered in two of four areas (Fig. 3). However, we contend that the most likely of these to have furnished the type specimen is the Mt Banahaw complex which apart from Mt Banahaw itself include the mountains of San Cristobal and Banahao de Lucuban.

Mt Banahaw is also spelled as Banhao, Banàhao, or as Banajao, and in the Spanish colonial period was known as as Monte de Majayjay or Monte San Cristobal. Of the 13

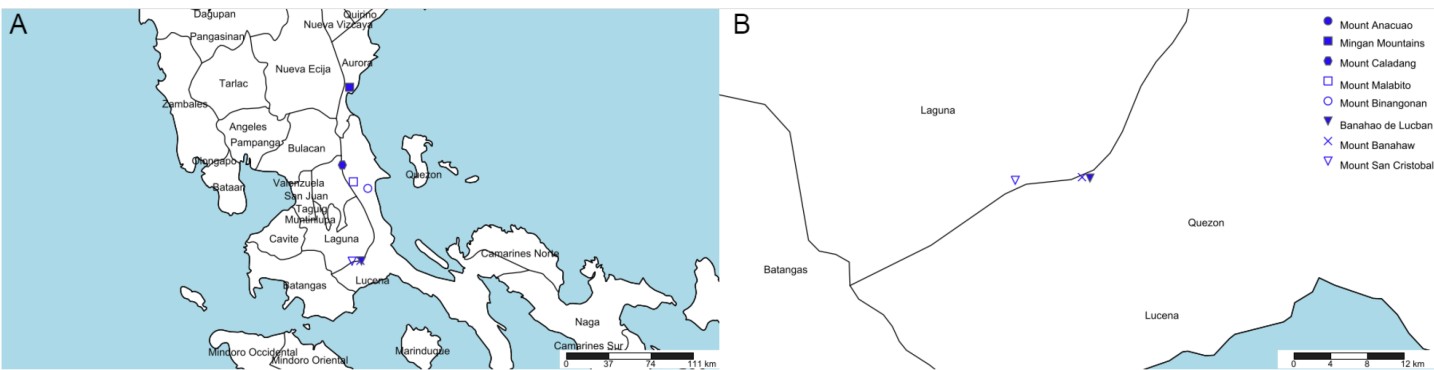

**Figure 3 Potential locations for *Nepenthes maximoides*.** (A) Locations of mountains above 1,000 m high in former Tayabas Province of Luzon (now Quezon and Aurora Provinces); (B) mountains above 1,000 m high found in Quezon Province. Drawn by Charles King.

specimens detailed on JStor as collected by H.M. Curran from Tayabas Province, six are given as from Mt Banahaw. The remaining seven specimens either have no further locality data or give Municipality Macuban or Paete-Piapi, lowland settlements to the NE of Mt Banahaw. Mount Banahaw lies on the boundary between Tayabas (now Quezon) and Laguna Provinces. Of the 42 H.M. Curran specimens recorded from Laguna Province on *JStor Global Plants (2020)*, 12 are also from Mt Banahaw. Therefore, on the basis of the JStor data, Mt Banahaw is the only collection location in former Tayabas Province that was a Curran location where *Nepenthes* might have been expected to be found (above 1000 m alt.). There is no evidence that any other mountain in Tayabas Province where *Nepenthes* might be found was visited by Curran, although this is possible. Moreover, Mt Banahaw was clearly a target for Curran since he collected there also while in Laguna Province. On the balance of probability, it seems likely, though not certain that the type and only specimen of *N. maximoides* derives from Mt Banahaw.

At the time the specimen was collected in 1911, the then provincial capital, Tayabas, was still a major administrative centre, giving its name to the former province. It is situated on the SE slopes of Mount Banahaw. The possible reasons for H.M. Curran selecting Banahaw are that it would then have been (1) the most readily accessible area of forest in the province from Curran's base near Manila, (2) it was and remains the largest block of surviving forest in the province and (3) he had visited it on a productive earlier visit: *Curran* 3039 collected with M.L. Merritt, is syntype of *Ahernia glandulosa* Merr. (Flacourtiaceae, now Achariaceae)—both a new species and genus, collected from 'Mt Banahaw, Tayabas Province' on 1 November 1907 (*JStor Global Plants, 2020*).

**Mount Banahaw**

At 2158 m high Mount Banahaw is the highest of a group of volcanoes south and east of Manila. Banahaw is flanked by the less high and more recent San Cristobal volcano on the west and Banahaw de Lucban on the NE. Andesitic-to-dacitic lava domes occur on the flanks of Banahaw and San Cristobal. The summit crater is about 2 km wide and 300 m deep. The last eruption is thought to have been in 1909 but this is uncertain

(*Global Volcano Program, 2020* continuously updated). Hot springs are present at several sites. The slopes of the mountain are completely forested apart from the lower altitudes which have largely been cleared from sea-level upwards to about 700 m alt. (see Conservation above). This complex contains c. 100 km$^2$ of forest above c. 500 m alt. There is no checklist of the plant species, but trees above 10 cm diameter at breast height have been characterised from 25 20 m × 20 m plots placed along a transect from 700 m alt. to the summit, which recorded 455 stems and 92 species (*Gascon et al., 2013*). Rainfall varies from 2350–2400 mm p.a. on the NW slope to 4470 m p.a. on the NE slope spread evenly over the year with 262 rainy days p.a. (*Gascon et al., 2013*). Details of the history of the protected status of the mountain are given under conservation (above). Currently it is part of the 10, 900 ha Mounts Banahaw–San Cristobal Protected Landscape. In describing *Rafflesia banahaw* Barcelona, Pelser & Cajano which is now considered a synonym of *R. philippensis* Blanco (*Barcelona et al., 2009*), *Barcelona, Pelser & Cajano (2007)* document nine botanists who made collections at Mount Banahaw, some who visited on more than one occasion. To these can be added the collector Azaola who obtained at this location on 22 April 1840 original material of *Rafflesia lagascae* Blanco (*Blanco, 1845*; *Pelser et al., 2013*) and who appears to have been the first collector of plant specimens at Mount Banahaw. Additional to these ten, *Van Steenis-Kruseman & Van Steenis (1950)* also list: W. Kerr (1805), C. Wilkes (1842), N.J. Andersson (1853), A. Marche (1880) and E. Langlassé (1895) as having visited Mt. Banahaw. So, it might be that Kerr rather than Azaola was the first western botanist to collect on the mountain. In this paper we add Curran (see above). Mount Banahaw is therefore one of the most botanically visited of locations in the Philippines. However, it is possible that important parts of the mountain, with intact vegetation, have yet to be thoroughly surveyed for plant species, and perhaps *Nepenthes maximoides* might be found in one of these places.

To the best of our knowledge, no other *Nepenthes* species is known from Mt. Banahaw.

### Recircumscription of Sect. Alatae

The *Nepenthes alata* group was first designated (*Cheek & Jebb, 2013a*) to include all species occurring from Luzon to Mindanao (excluding Palawan) excepting those of Sect. *Insignes* Danser, and also excluding the only Philippine non-endemic species, *N. mirabilis* (Lour.) Druce.

Subsequently the *N. micramphora* group was designated (*Cheek & Jebb, 2013b*) for three species from Mindanao that lacked key attributes of the *N. alata* group as then defined—particularly the basal lid appendage and a distinct petiole. These two groups were later formalised as Sect. *Alatae* Cheek & Jebb and Sect. *Micramphorae* Cheek & Jebb respectively (*Cheek & Jebb, 2015*).

However, a recent near-comprehensive species-level phylogenomic study of *Nepenthes* revealed that Sect. *Alatae* was not monophyletic (*Murphy et al., 2020*). The northern Sect. *Alatae* species, that is of Luzon and Sibuyan, are sister to the species of Palawan, previously not considered to be closely related. Unexpectedly, the southern *Alatae* species that is

of the southern Visayas (Negros, Leyte and Mindanao (excluding *N. graciliflora* Elmer which extends from Luzon)), including the *Micramphorae*, are sister to the Palawan clade & the northern *Alatae*. The division of the northern from the southern *Alatae* revealed by phylogenomic analysis is supported by morphology as seen in the primary division in the most recent key to the group (*Cheek & Jebb, 2014*). The species of the north have monomorphic, uniformly small, moderately crater-like nectar glands evenly spread on the lower surface of the lid. In contrast, the species of the south usually have dimorphic glands in different size-classes and always have at least some glands very much larger than those seen in the northern species. While the northern species have 1-flowered partial-peduncles, those of the south have 2-flowered partial peduncles (where known).

The separation between the two redelimited sections can be summarised in key form as follows:

Lower surface of lid, including basal appendage (if present), densely and evenly covered in uniformly minute circular or shortly elliptic nectar glands (0.15–0.25(–0.3) mm diam.); inflorescence partial-peduncles 1-flowered. Luzon & Sibuyan (except *N. graciliflora* Luzon to Mindanao) . . . . . . . . . . . . . . . . . . . . . . . . . . . . . . . . . . . . . . . . . . . . . .Sect. **Alatae**

Lower surface of lid with nectar glands either absent from the appendage and/or, sparse, large or dimorphic (larger glands 0.35–0.4 mm diam. or larger); inflorescence partial-peduncles 2-flowered. Southern Visayas & Mindanao. . . . . . Sect. **Micramphorae**

*Nepenthes argentii* Jebb & Cheek of Sibuyan was formerly unplaced in a species group (*Cheek & Jebb, 2001*). Due to the shortly cylindrical pitchers, a broad peristome extending as a column to the lower lid surface and with blade-like peristome ridges it has similarities with species of the Palawan group. But it is embedded in the northern *Alatae* clade on phylogenomic data (*Murphy et al., 2020*). This unexpected placement is morphologically supported by its lid nectar gland and inflorescence morphology: it has uniform, minute nectar glands on the lower surface of the lid and 1-flowered partial-peduncles. Since the type species of Sect. *Alatae* is *N. alata* Blanco of Luzon, the northern species must retain this sectional name. The only available sectional name for the southern species of Sect. Alatae is Sect. *Micramphorae*, so this name must perforce now be adopted for the 'southern Alatae' which comprise the most species-diverse of the *Nepenthes* clades of the Philippines. A key to Sect. *Alatae sensu stricto* (as here recircumscribed), informally referred to as the 'northern Alatae' above, is presented in the results (above). It now consists of six species, all but one of which occur on Luzon, with *N. graciliflora* extending from Luzon to Mindanao, and *N. armin* being restricted to Sibuyan.

### The apical lid appendage in *Nepenthes*

*Nepenthes maximoides* is highly remarkable and among all the known species of the genus in the Philippines unique, in its filiform apical appendage (Fig. 2G). This structure also occurs in several species of sect. *Regiae* (Borneo to New Guinea), where it appears homologous—the appendage appears to be a continuation of the distal terminus of the midline of the lid. This midline is thickened, raised above the adjoining tissue as a low

ridge, and appears to be highly vascularised. In Sect. *Regiae* the apical appendage often carries the largest nectar glands present on the lower surface of the lid (*Cheek & Jebb, 2001*), and is also often raised above the surface as a ridge for part of its length: see for example *Nepenthes minima Cheek & Jebb (2016b)* Lid appendages also occur in some species of sect. *Montanae* Danser, but these appear likely to be non-homologous, arising differently, as lobes from the midline far before its distal terminus for example *N. lingulata* Chi C. *Lee, Hernawati & Akhriadi (2006)*.

The earliest clade in the most comprehensive phylogenetic tree of *Nepenthes* (*Murphy et al., 2020*) in which an apical lid appendage is developed is in the *Nepenthes danseri* group, as seen in *N. weda* Cheek (*Cheek, 2015*). In this species all stages of the pitchers were available for study and there appears to be stage dependent heteromorphy of the distal lid appendage. In rosette pitchers the appendage is a transversely crescent-shaped ridge, while in upper pitchers it protrudes further from the surface and is more elaborate but not filiform (*Cheek, 2015*: 225). While potentially proto-filiform appendages such as these are unknown in Sect. *Alatae* as now delimited, they do occur in Sect. *Micramphorae* in *N. robcantleyi* Cheek (*Cheek, 2011*) and in *N. tboli Cheek & Jebb (2014)*.

**Convergence of pitcher morphology**
The shared pitcher morphology of *Nepenthes maximoides* with that of *Nepenthes maxima* appears due to convergence, not due to phylogenetic proximity. Convergence of pitcher morphology is recorded in other cases in the genus (*Thorogood, Bauer & Hiscock, 2018*). Recently, 12 functional pitcher types have been recognised, each postulated to target capture of nutrients from animals (sometimes plants) in a different manner (*Cheek, Jebb & Murphy, 2020*). *Nepenthes maxima* and *Nepenthes maximoides* fall under pitcher type 2 ('narrow-funnel' *Cheek, Jebb & Murphy, 2020*).

## CONCLUSIONS
The dramatic rise in the numbers of Philippine species of *Nepenthes* in the 21st century (see "Introduction") is mirrored in other plant groups such as *Rafflesia* R.Br. (Rafflesiaceae). Before 2002 only two species of *Rafflesia* were thought to be known from the Philippines (subsequently two additional, long-overlooked species came to light), and, as in *Nepenthes*, the genus was thought to be most diverse in Borneo and Sumatra. Intensive fieldwork in remaining patches of forest in the Philippines, however, has raised species numbers steadily from two species in 2002 to 13 species in 2019, and Philippines now is the most species-diverse country for *Rafflesia* globally (*Barcelona, Pelser & Cajano, 2007*; *Barcelona et al., 2009*; *Pelser et al., 2013*, *2019*).

The number of flowering plant species known to science is disputed (*Nic Lughadha, Bachman & Govaerts, 2017*), but a reasonable estimate is 369,000 (*Nic Lughadha et al., 2016*), while the number of species described as new to science has been at about 2,000 per annum for at least 10 years (*Cheek et al., 2020*). The conservation status of 21–26% of plant species has been established using evidence-based assessments, and 30–44% of these rate the species assessed as threatened, while only c. 5% of plant species have been assessed using the *IUCN (2012)* standard (*Bachman, Nic Lughadha & Rivers, 2018*).

Newly discovered species such as *Nepenthes maximoides*, are likely to be threatened, since widespread species tend to have been already discovered and it is the more localised, rarer species that remain to be found although there are exceptions such as *Gouania longipedunculata Cahen, Stenn & Utteridge (2020)* which is widespread. This makes it urgent to discover and protect such localised species before they become extinct due to habitat clearance as was the case with *Nepenthes extincta Cheek & Jebb (2013a)*. However, it may be too late for *Nepenthes maximoides*, which may be extinct already, although efforts to rediscover it should be made in case not.

## ACKNOWLEDGEMENTS

The authors thank Dr. Laurence Dorr (US) for facilitating the study visit of Martin Cheek to PH (at the Academy of Natural Sciences of Drexel University, Philadelphia, incorporating the Pennsylvania State University herbarium) and to that Institute for providing on loan to K the material for the basis for the description in this article of *Nepenthes maximoides*. Janis Shillito typed up much of the manuscript. We give praise and thanks to two reviewers Pieter Pelser and Victor Amoroso for their painstaking, valuable and constructive reviews of an earlier version of this article.

We thank the staff at the Tropical Nursery, RBG Kew for cultivating so expertly over many years the species *Nepenthes* collection used in this paper for comparison purposes, especially Tom Pickering, Rebecca Hilgenhof, Nick Johnson, James Beattie, Lara Jewitt and Kath King.

### Funding

The authors received no funding for this work.

### Competing Interests

The authors declare that they have no competing interests.

### Author Contributions

- Charles King performed the experiments, analyzed the data, prepared figures and/or tables, and approved the final draft.
- Martin Cheek conceived and designed the experiments, performed the experiments, analyzed the data, prepared figures and/or tables, authored or reviewed drafts of the paper, and approved the final draft.

### Data Availability

All data is available in the article. The basis of this paper is a specimen which is featured as Fig. 1. The specimen is *Curran s.n.*

Accession numbers are: Herb. Univ. Pennsylvania sheet number 70707, Academy of Natural Sciences Philadelphia sheet number 01113309 (holotype PH) as stated in the

results where full data (description) is given. This herbarium is currently included in what is now the Academy of Natural Sciences of Drexel University, Philadelphia.

## New Species Registration

The following information was supplied regarding the registration of a newly described species:

Nepenthes maximoides Cheek: 77211103-1.

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
