# Peer review of "Nepenthes maximoides (Nepenthaceae) a new, critically endangered (possibly extinct) species in Sect. Alatae from Luzon, Philippines showing striking pitcher convergence with N. maxima (Sect. Regiae) of Indonesia"

_PeerJ, doi:10.7717/peerj.9899_

## Round 0.1 · original submission · Minor Revisions

Many thanks for submitting your manuscript to PeerJ describing a new species of Nepenthes. I have now reviewed the manuscript and receive 2 reviews that are both supportive. They have provided some detailed points that may help improve and provide clarity particularly around comparison of this new species with other. I therefore recommend minor revisions and look forward to seeing the revised manuscript and the detail rebuttal letter.

·

Basic reporting

• The manuscript is well written and clear. I have pointed out a few minor typing errors below.
• The manuscript is contains sufficient background information (although see a few minor comments below) and contains appropriate literature references (but see below).
• The authors wrote in the first paragraph of the Introduction that 12 species of Nepenthes were published in 2013 and provide literature references to these. Although, I think there were indeed 12 new species described in that year, the references do not include a reference to the publication of Nepenthes viridis Micheler, Gronem., Wistuba, Marwinski, W.Suarez & V.B.Amoroso. I suggest to add this.
• I suggest to include a photo of the type specimen with a close-up of the herbarium label. I think that this would be informative, because this specimen is discussed in quite a bit of detail in the Introduction and Discussion. If available, it would also be nice if the authors could present photographs or an illustration of N. maxima, so that the readers can compare the morphology of the new species with that of the latter.
• I suggest to show the character states in which N. maximoides is different from N. maxima (as mentioned in the third paragraph of the Introduction) more clearly in Figure 1: close-ups showing 1) the patent wings of the petioles, and 2) an axillary bud. Fig. 1K is not labelled as such (i.e. ‘K’ is missing in the figure). Scales for Fig. 1C and 1E appear to be missing. Fig. 1B is a bit too small to make out enough detail regarding the spur, such as its bifid apex.
• Line 138: Can the authors make it clearer which part of the plant they are referring to when they write “the indumentum is glabrous [Should this be ‘absent’, btw?] or inconspicuous? Perhaps this can also be illustrated in Fig. 1.
• Lines 142-148: “The shared pitcher…”. This might be better placed in the Discussion, where also the homology/homoplasy of other morphological characters is discussed.
• Line 152: ‘We’ instead of ‘we’.
• Line 156: “The methods used in this paper are as those documented in Cheek & Jebb (2013a-g).” Considering that a detailed methodology is provided in lines 176-181, I wonder if this sentence is necessary. Does it refer to methods not mentioned here? If so, could these be concisely presented?
• Line 198: Replace “maximioides” with ‘maximoides’.
• Couplet 4 in the identification key: Can the authors make it clearer that the text about the “outer surface” refers to the outer surface of the upper pitchers? Perhaps this can be done by replacing the second semi-colon in lines 207 and 210 with a comma?
• Please check that the character states that are mentioned in the various parts of the manuscript are described in the same way. For example, the upper pitchers of N. maximoides are described as “infundibulate” in line 197, but “infundibulate-cylindrical” in line 222. I have not looked for other such inconsistencies.
• In the diagnosis, the authors compare N. maximoides with N. graciliflora. Can they make it more explicit why they chose the latter species instead of another species in Sect. Alatae? Is it the one that is most similar to N. maximoides?
• Line 225/226 “(not symmetrical)”? It is not clear to me how this contrasts with “the lid with a hooked basal appendage”.
• Line 229: What does “st.” mean?
• Line 234: Should “(probably)” be placed before instead of after the first comma in this line?
• Line 338: Add space between “IPNI” and “(continuously updated)”.
• Line 367: “where none are known to survive”. This contrasts with what is written in van Steenis-Kruseman, M.J, & van Steenis, C.G.G.J. (1950). Malaysian plant collectors and collections being a Cyclopaedia of botanical exploration in Malaysia and a guide to the concerned literature up to the year 1950. Flora Malesiana - Series 1, Spermatophyta, 1(1), 2–639, which mentions “Many plants in Herb. Manila, at least partially numbered in the F.B. series” (p. 123/124). I also recommend that the authors refer to this publication and, where desired, amend their section in the Discussion about H.M. Curran. Note, for instance that his middle name is spelled as McCullom in this publication.
• Line 429-434: In addition to the names listed in this manuscript and in Barcelona et al. (2007), Van Steenis-Kruseman & Van Steenis (1950) also list: W. Kerr (1805), C. Wilkes (1842), N.J. Andersson (1853), A. Marche (1880) and E. Langlassé (1895) as having visited Mt. Banahaw. So, it might be that Kerr rather than Azaola might have been the first western botanist to collect on the mountain. I suggest to amend the text accordingly.
• Line 443: Add period after “Druce”
• Line 445: Replace “were” with “was”
• Line 463: Replace “partial-peduncle” with ‘partial-peduncles’.
• Please check that the full initials of the authors are included in your reference list. For example, "Barcelona J, Pelser P, Cajano M.” should be 'Barcelona JF, Pelser PB, Cajano MO.’

Experimental design

• The authors present new and original scientific knowledge and used a suitable and well-established methodology to determine if the specimen collected by Curran represents a new species. This methodology is described in sufficient detail and the results are clearly presented.

Validity of the findings

• This manuscript presents convincing evidence that N. maximoides is an undescribed species.
• Line 142: “It seems clear that Curran s.n. is correctly labelled as a Philippine species, after all”. This cannot be concluded from the preceding, because this may still have been the case and this possibility cannot be excluded. Perhaps the authors could write “It seems clear that Curran s.n. might have been correctly labelled as a Philippine species, after all”.

Additional comments

• "Before 2002 only two species of Rafflesia were known from the Philippines…” Although this was previously believed to be the case, we now know that in addition to R. manillana and R. schadenbergiana, also R. lagascae and R. philippensis were described before 2002. However, both were in 2000 considered to be synonyms of R. manillana. This should be made clear in the manuscript.
• Perhaps the authors can mention if any other Nepenthes species have been collected from Mt. Banahaw? This might be relevant as context.

·

Basic reporting

1. clear and exhaustive literature but with minor comments:
Line 152 2nd word capitalize We

Experimental design

methods described with sufficient detail and information

L176 & L297- why not use most recent IUCN citation?

Validity of the findings

1. all underlying data has been provided and robust

2. conclusions are scientifically stated

3. L198 - N. maximioides=N. maximoides

4. L304-306- Though the geographical origin of Luzon and Mindanao is different, it is possible that this species is also found in Southern Philippines. I suggest that you compare this with N. copelandii which is found Southern Philippines. We have specimens of this species with upper and lower/intermediate pitchers.
5. L284 Luban=Lucban
6. L386- delete "as"

Additional comments

The following are my comments and suggestions:

1.) Add keywords

2.) Add table comparing N. maximoides with morphologically related species from section Alatae i.e. N. copelandii, the upper pitchers of this species looks similar with the upper pitchers of N. maximoides. Moreover, this species is also found at about 1,000 m asl in Mt. Apo.

3.) In the taxonomic key if possible add other Philippine Nepenthes species from section Alatae

4. Why is N. maximoides compared with N. graciliflora when it is not the closest (morphologically similar) species? I suggest that the new species, N. maximoides should be compared with the closest species belonging to Section Alatae by indicating its distinct characters

---

## Round 0.2 · accepted · Accept

Many thanks for submitting your manuscript to PeerJ and the careful consideration of the comments, in particular including the comparison with N. copelandii which strengthened the paper. I am therefore delighted to recommend that the paper be accepted.